# Factors related to subjective well-being among community-dwelling older adults living alone: A stratified analysis by sex and marital status from the JAGES

Nana Abe[1], Nanami Oe[2], Etsuko Tadaka[2]*, Toshiyuki Ojima[3]

**1** Sapporo City Public Health Center, Sapporo, Hokkaido, Japan, **2** Faculty of Medicine/Graduate School of Health Sciences, Department of Community and Public Health Nursing, Hokkaido University, Sapporo, Hokkaido, Japan, **3** Department of Community Health & Preventive Medicine, Hamamatsu University School of Medicine, Hamamatsu, Shizuoka, Japan

* e_tadaka@pop.med.hokudai.ac.jp

## Abstract

### Background

Previous cross-sectional studies suggest that negative health outcomes such as mortality, social isolation, loneliness, and depression among older adults living alone vary by sex and marital status, with men often worse off than women and unmarried people worse off than married people. However, limited evidence exists from longitudinal studies regarding whether positive health outcomes such as subjective well-being (SWB) also vary by sex and marital status. The focus by sex and marital status on the positive health outcomes and diverse profiles of older adults living alone is important for public health in the near future. Therefore, the purpose of this study was to identify changes in SWB over time and its associated factors by sex and marital status among older adults living alone in the community using a longitudinal study in a representative population.

### Methods

This was a longitudinal study using data from the Japan Gerontological Evaluation Study. This study is the first to reveal differences in SWB and related factors over 3 years among older adults living alone in the community (n = 8,579) who were stratified by sex and marital status (married men, non-married men, married women, and non-married women).

### Results

Women moved to higher levels of SWB than did men, and married individuals moved to higher levels of SWB than did unmarried individuals. Independent functioning factors and interpersonal factors were significantly associated with SWB for married men and married women, but for unmarried women, the association by interpersonal factors was more pronounced, and for unmarried men, only limited emotional support and health promotion activities were significant among the interpersonal factors.

**Data Availability Statement:** All data are from the Japan Agency for Gerontological Evaluation Study (URL: https://www.jages.net/). Data cannot be shared publicly because all JAGES datasets have

ethical or legal restrictions for public deposition due to inclusion of sensitive information from the human participants. All inquiries on the data are to be addressed at the data management committee via e-mail: dataadmin.ml@jages.net.

**Funding:** This study used data from JAGES (the Japan Gerontological Evaluation Study). This study was supported by Grantin-Aid for Scientific Research (15H01972, 15H04781, 15H05059, 15K03417, 15K03982, 15K16181, 15K17232, 15K18174, 15K19241, 15K21266, 15KT0007, 15KT0097, 16H05556, 16K09122, 16K00913, 16K02025, 16K12964, 16K13443, 16K16295, 16K16595, 16K16633, 16K17256, 16K17281, 16K19247, 16K19267, 16K21461, 16K21465, 16KT0014, 17K04305, 17K04306, 25253052, 25713027, 26285138, 26460828, 26780328, 18H03018, 18H04071, 18H03047, 18H00953, 18H00955, 18KK0057, 19H03901, 19H03915, 19H03860, 19K04785, 19K10641, 19K11657, 19K19818, 19K19455, 19K24060, 19K20909, 20H00557, 21K19635, 21H03153) from JSPS (Japan Society for the Promotion of Science); Health Labour Sciences Research Grants (H26-Choju-Ippan-006, H27-Ninchisyou-Ippan-001, H28-Choju-Ippan-002, H28-Ninchisyou-Ippan-002, H30-Kenki-Ippan-006,H29-Chikyukibo-Ippan-001, H30-Jyunkankinado-Ippan-004, 19FA1012, 19FA2001, 21FA1012, 22FA2001,22FA1010), Research project on health and welfare promotion for the elderly from the Ministry of Health, Labour and Welfare, Japan; the Research and Development Grants for Longevity Science from Japan Agency for Medical Research and development (AMED) (JP18dk0110027, JP18ls0110002, JP18le0110009, JP20dk0110034, JP21lk0310073, JP21dk0110037, JP22lk0310087), the Research Funding for Longevity Sciences from National Center for Geriatrics and Gerontology (24-17, 24-23, 29-42, 30-30, 30-22, 20-19, 21-20); Open Innovation Platform with Enterprises, Research Institute and Academia (OPERA, JPMJOP1831) from the Japan Science and Technology (JST); a grant from the Japan Foundation For Aging And Health (J09KF00804), a grant from Innovative Research Program on Suicide Countermeasures (1-4), a grant from Sasakawa Sports Foundation, a grant from Japan Health Promotion & Fitness Foundation, a grant from Chiba 42 Foundation for Health Promotion & Disease Prevention, the 8020 Research Grant for fiscal 2019 from the 8020 Promotion Foundation (adopted number: 19-2-06), grants from Meiji Yasuda Life Foundation of Health and Welfare. The views and opinions expressed in this article are those of the authors and do not necessarily reflect the official policy or position of

## Conclusions

This study revealed that among older adults living alone, changes in SWB over time and the independent functioning factors and interpersonal factors associated with this change varied by sex and marital status among older people living alone. These findings are useful for policy-making and guiding intervention activities to promote SWB in a society in which the environment for older adults living alone is changing dramatically.

## Introduction

Approximately one-quarter of older adults aged 60 years and older in the United States, Canada, and Italy and about one-third of this cohort in Germany, France, and the United Kingdom live alone [1]. In Japan, 50.1% of households were three-generation households and only 10.7% were living-alone households in 1980; by 2019, three-generation households had declined to 9.4% and more than one-quarter (28.8%) of all households were living-alone households [2]. Furthermore, not only has the number of people aged 65 years and over living alone increased approximately seven-fold from 1980 to 2020, but the percentage of this population has more than doubled from 4.3% to 15.0% for men and 11.2% to 22.1% for women [2]. This indicates that the household environment for older adults is changing dramatically. Previous studies have shown that living alone is a risk factor for a variety of health outcomes including mortality [3,4], social isolation[5] loneliness [6,7], depression [7], and frailty [8] and that these factors are associated with sex differences. These findings indicate the need to focus attention on older adults living alone in Japan and other developed countries.

The increase in the population of older adults living alone is attributable to changes in marital status. In Japan, the number of adults who were unmarried at age 50 years increased from 3.9% to 28.3% of men and from 4.3% to 17.8% of women between 1985 and 2020 [9]. This trend is expected to continue in the future, with older adults expected to continue living alone into old age. Unmarried individuals—and especially never-married persons—have a higher mortality rate than married individuals and among unmarried individuals, men have a higher mortality rate than women [10]. Other studies have further reported mortality rates by marital status, showing higher mortality among divorced individuals and males over 65 years of age [11]. Although marital status was not a direct factor in mortality, these studies suggest that long-term marital status may be associated with differing effects on health. In addition, studies examining health outcomes and marital status have described lower self-reported health, lower life satisfaction, and a higher likelihood of depression in those who were not married compared with those who remained married [12]. For these reasons, focusing on marital status and sex to accurately understand the health outcomes of older community-dwelling adults is important.

As described above, although negative health outcomes and associated factors are known to differ by sex and marital status, positive health outcomes—particularly the key outcome of subjective well-being (SWB) that is associated with public health and health longevity aspects —and its related factors are not clear by sex and marital status. Subjective well-being can be expressed as a "person believes or feels that his or her life is going well" [13]. Subjective well-being has attracted attention as a health indicator and is one of the factors incorporated into the model framework of successful aging [14–16]. Subjective well-being is associated not only with elements of socioeconomic status such as sex, marital status, educational attainment, and household income [13], but also with depression [17,18] and mortality [19]. Factors specific to

the respective funding organizations. The funders had no role in study design, data collection and analysis, decision to publish, or preparation of the manuscript.

**Competing interests:** The authors have declared that no competing interests exist.

older adults' activities or external interventions that are associated with SWB include exercise, hobbies, knowledge of nutritional balance, and participation in community and volunteer activities [20]; however, marital status has not been investigated as a factor. In previous studies of the relationship between marital status and SWB, the variation in SWB attributable to marriage-related events including divorce and widowhood has been mentioned [13]; however, the never-married group has not received much attention. Too few longitudinal studies of representative populations fully account for non-married persons and sex differences to allow causal inferences about SWB and related factors.

The number of older adults living alone, which is expected to increase in Japan and other developed countries, is likely to be diverse with regard to sex and marital status. Therefore, focusing on the four population categories of older adults living alone based on sex and marital status (married men, unmarried men, married women and unmarried women)—rather than on the overall group of older adults living alone—while also focusing on not only negative health outcomes but also positive health outcomes such as SWB is necessary. Moreover, studying representative populations in longitudinal studies from which causal relationships between SWB over time and their association can be inferred is critical for developing evidence-based health policy that is responsive to new populations and diversity. Therefore, in this study, by using a longitudinal study of a representative population (the Japan Agency for Gerontological Evaluation Study), our aim was to determine changes in SWB over time and their association with independent functioning factors and interpersonal factors by sex and marital status (married men, unmarried men, married women and unmarried women) among older adults living alone in the community. Independent functioning factors are critical for performing daily living tasks without help [21,22]. The achievement of functional independence ensures that older adults can participate fully in meaningful and purposeful life events—a requirement for SWB. Interpersonal factors are individual elements or a group of factors that positively or negatively influence the quality of relationships [21,23] and are integral to creating and maintaining meaningful personal relationships that ensure SWB among older adults in the community. Our hypotheses were as follows:

Hypothesis 1: SWB is more likely to transition to a high level in women rather than men and in married versus unmarried individuals among community-dwelling older adults living alone.

Hypothesis 2: Independent functioning factors and interpersonal factors are significantly associated with SWB in older adults living alone and differ by sex and marital status among community-dwelling older adults living alone.

## Methods

### Study population

In this prospective longitudinal study, we used data from two waves (2016 and 2019) of the Japan Agency for Gerontological Evaluation Study (JAGES), Japan's first large-scale database project regarding older adults aged 65 and over. The JAGES includes Japanese people aged 65 or older who are not eligible for public long-term care insurance benefits. Survey participants were selected in each municipality (city, town, and village) with random sampling in large municipalities ($\geq$ 5,000 people) and inclusion of all participants in small municipalities ($<$ 5,000 people). The baseline survey was conducted between September 2016 and January 2017 and covered 39 municipalities (34 insurers) in 18 prefectures. The follow-up survey was conducted between November 2019 and January 2020 for 63 municipalities (56 insurers) in 18 prefectures. In the first and second waves, questionnaires were collected from 196,438 and

**Table 1.** The study participants living alone who answer sex, marital status and SWB.

| | Baseline 2016[a] (n = 25,518) | | Follow up 2019[b] (n = 28,664) | | Baseline–Follow up 2016–2019 (n = 8,579) | |
|---|---|---|---|---|---|---|
| | n | (%) | n | (%) | n | (%) |
| Married Men | 5,868 | (23.0) | 6,835 | (23.8) | 1,803 | (21.0) |
| Unmarried Men | 1,925 | (7.5) | 2,592 | (9.0) | 709 | (8.3) |
| Married Women | 15,905 | (62.3) | 17,323 | (60.4) | 5,372 | (62.6) |
| Unmarried Women | 1,820 | (7.1) | 1,914 | (6.7) | 695 | (8.1) |

[a]The baseline survey was conducted between September 2016 and January 2017.

[b]The follow-up survey was conducted between November 2019 and January 2020.

261,428 participants, respectively, with response rates of 70.2% and 69.4%, respectively. The target populations were older adults living alone, with 275,69 cases in 2016 and 30,958 cases in 2019. The exclusion criteria were: (1) those missing information regarding sex, SWB, or marital status, (2) those who reported "other" marital status, and (3) those whose marital status changed between the first and second waves. The numbers of eligible cases were 25,518 and 28,664 in 2016 and 2019, respectively. The 28,664 participants at follow-up included 16,936 participants who were involved in the survey for the first time in 2019 and 11,728 participants who had participated in the survey since 2010, 2013, or 2016. Ultimately, the analysis included 8,579 respondents who provided responses at the two time points (2016 and 2019; Table 1).

## Social demographics

Social demographics included age, sex, marital status, education level, and annual household income. We categorized marital status into two groups: married (married, widowed, and divorced) and unmarried (never married). In this study, we stratified participants into four groups according to sex and marital status: married men, unmarried men, married women, and unmarried women. The married men group consisted of 1,803 men, among whom 63 (3.5%) were currently married, 1,005 (55.7%) were widowed, and 735 (40.8%) were divorced. The married women group consisted of 5,372 women, among whom 54 (1.0%) were married, 4,319 (80.4%) were widowed, and 999 (18.6%) were divorced. In addition, individuals in the "married with a spouse" group were assumed to be those whose spouses lived alone because they were moved to an institution or for other reasons.

## Dependent variable

**Subjective well-being.** We measured SWB with the following question [21]: "How happy are you currently? Please provide a score on a 10-point scale from 0 (very unhappy) to 10 (very happy)." We categorized the SWB score as high level and low level. High-level SWB was defined as a median SWB score of 7–10 points in men and 8–10 points in women; low-level SWB was defined as a median SWB score of 1–6 points in men and 1–7 points in women.

## Independent variable

**Independent functional factors.** Lifestyle variables included the frequency of eating meat and fish, the frequency of eating vegetables and fruits, alcohol consumption, and smoking status. Participants chose from seven categories of eating frequency: "at least twice daily," "once a day,", "4–6 times a week," "2–3 times a week," "once a week," "less than once a week," and "not eating." The seven categories were then recategorized into three categories: "less than

once a week," "2–6 times a week", and "eat every day." The following categories were used in JAGES for alcohol consumption: "currently drink," "I stopped drinking within the last five years and don't drink now," "I stopped drinking more than five years ago and don't drink now," and "I have never been a drinker." In this study, we recategorized alcohol consumption into three categories: "never been a drinker," "drank in the past," and "currently drink." The following options were used in JAGES to categorize smoking status: "I smoke almost every day," "I smoke sometimes," "I stopped smoking within the last five years and don't smoke now," "I stopped smoking more than five years ago and don't smoke now," and "I have never been a smoker." In this study, we recategorized smoking status into three categories: "never been a smoker," "smoked in the past," and "currently smoke."

Self-reported health was measured on a 4-point Likert scale using the question "How is your current health condition?"

The Instrumental Activities of Daily Living (IADL) were measured using the Tokyo Metropolitan Institute of Gerontology Index of Competence (TMIG-IC). This scale is a 13-item multidimensional scale measuring higher-level competence in older adults [22]. Each item is scored as 1 point for "Yes" and 0 points for "No;" a higher score is considered a higher-level IADL. The Tokyo Metropolitan Institute of Gerontology Index of Competence consists of three sub-levels: Instrumental Self-Maintenance (range: 1–5 points), Intellectual Activity (range: 1–4 points), and Social Role (range: 1–4 points). As a response to the Instrumental Self-Maintenance question in the JAGES survey form, participants were asked to choose between "I can," "I can but I haven't," and "I can't." We scored "I can" and "I can but I haven't" as 1 point and "I can't" as 0 points [23,24].

The Geriatric Depression Scale (GDS) is a tool for the evaluation of depression in older adults [25]. The JAGES survey used the GSD–15, a short version of the GDS. The GDS-15 is an effective screening instrument for major depression and is easy to answer for older adults [26]. The scale ranges from 0 to 15, with a high score indicating severe depression.

**Interpersonal relationships factors.** The frequency of meeting friends or acquaintances was classified into six categories: "no contact," "a few times a year," "1–3 times a month," "once a week," "2–3 times a week," and "at least 4 times a week." Relationships with neighbors were measured using the following question: "What kind of relationship do you have with your neighbors in your community?" Participants were given the following choices: "I cooperate with neighbors daily, such as consulting with each other or lending and borrowing daily necessities," "I have a daily relationship with neighbors to the extent of chatting with them," "I have no more than exchanged greetings with neighbors," and "I don't socialize with neighbors."

We used the receipt and provision of emotional and instrumental support as social support variables. Receiving emotional support was assessed using the following question: "Do you have someone who listens to your concerns and complaints? (Select all that apply)." The provision of emotional support was assessed with the statement: "Do you listen to someone's concerns or complaints? (Select all that apply)." The receipt of instrumental support was assessed using the following question: "Do you have someone who can take care of you when you are sick in bed for a few days? (Select all that apply)." The provision of instrumental support was assessed with the statement: "Do you take care of someone when they are sick in bed for a few days? (Select all that apply)." The person providing or receiving support was selected from the following options: spouses; children living together, children living separately, siblings or other relatives, grandchildren, neighbors, friends, others, or no such person. We counted the type of person who provided or received emotional or instrumental support.

The JAGES survey asked participants how often they join in each of eight types of social participation groups or clubs: volunteer groups, sports groups or clubs, hobby groups, senior

citizens' clubs, neighborhood associations, learning or cultural groups, health promotion activities, and intergenerational exchange. Participants were allowed to choose the following answers: "at least 4 times a week," "2–3 times a week," "once a week," "1–3 times a month," "a few times a year," and "do not participate." In this study, we recategorized these groups into "non-participation" and "participation" and counted the types of social participation.

## Statistical analyses

Descriptive statistics for demographic and independent variables were presented by sex and marital status. Change in SWB level from 2016 to 2019 was described. A logistic regression analysis was performed with the dependent variables "1" for high-level SWB in 2019 and "0" for low-level SWB in 2019. Multivariate analyses were not performed for each independent variable, because of multicollinearity. Univariate logistic regression analyses were performed for each independent variable, and in each univariate logistic regression analysis, gender and SWB level at baseline were entered and adjusted to determine whether the 3-year change in SWB was associated with independent functional factors and interpersonal relationship factors, adjusting for age and SWB level at baseline. IBM SPSS (ver. 29.0; IBM Corp., Armonk, NY, USA) was used for all statistical analyses and a P value of less than 0.05 was considered to be statistically significant.

## Ethical statement

All the participants received a written explanation from their member JAGES municipality and provided their consent in writing when responding to the survey. This research was conducted in accordance with the 1964 Declaration of Helsinki (and its amendments), and the ethical guidelines for life sciences and medical research involving human subjects presented by the Ministry of Health, Labour and Welfare of Japan. This study was reviewed and approved by the Institutional Review Board of Hokkaido University (Certificated No, 22–2; 6/May/2022) and the Institutional Review Board of JAGES (Certificated No, J-AGES5760; 1/Aug/2022).

## Results

Table 2 shows the baseline demographic characteristics of participants by sex and marital status. The mean age of all participants at baseline was 74.1 ± 5.7 years. The highest mean age in the four groups was in married women (74.8 ± 5.6 years) and the lowest was in unmarried men (69.9 ± 3.9 years). The highest mean SWB score in the four groups was in married women (7.3 ± 1.9 points) and the lowest was in unmarried men (5.6 ± 1.8 points). Both overall and in each sub-category of sex and marital status, participants most commonly had 10–12 years of education and an annual household income of 1.50–2.99 million yen.

Table 3 shows the independent functional and interpersonal relationship factors by sex and marital status. Women ate meat, fish, vegetables, and fruits more frequently than men. Moreover, men more frequently responded with "currently drink" and "currently smoke" than women. The frequency of contact with friends and the intensity of relationships with neighbors were higher in the married group than in the unmarried group for both men and women. Men more frequently received and provided emotional and instrumental support than women. Married women were involved in most types of social participation.

Table 4 illustrates the transition of SWB from baseline (2016) to follow-up (2019) according to sex and marital status. The proportions of participants who transitioned to the high-level SWB group were 44.7%, 31.0%, 49.3%, and 41.2% in married men, unmarried men, married women, and unmarried women, respectively.

**Table 2. Baseline characteristics of participants by sex and marital status.**

|  | Married Men | | Unmarried Men | | Married Women | | Unmarried Women | | All | |
|---|---|---|---|---|---|---|---|---|---|---|
|  | n = | 1,803 | n = | 709 | n = | 5,372 | n = | 695 | n = | 8,579 |
| Age, mean±SD(years) | 74.1 | ±6.1 | 69.9 | ±3.9 | 74.8 | ±5.6 | 72.9 | ±5.5 | 74.1 | ±5.7 |
| SWB, mean±SD(score) | 6.1 | ±2.0 | 5.6 | ±1.8 | 7.3 | ±1.9 | 6.9 | ±1.9 | 6.9 | ±2.0 |
| **Educational level, n(%)** | | | | | | | | | | |
| 6 years< | 15 | (0.8) | 6 | (0.8) | 31 | (0.6) | 1 | (0.1) | 53 | (0.6) |
| 6–9 years | 474 | (26.4) | 229 | (32.3) | 1,634 | (30.6) | 124 | (17.9) | 2,461 | (28.8) |
| 10–12 years | 703 | (39.1) | 283 | (40.0) | 2,486 | (46.6) | 310 | (44.8) | 3,782 | (44.3) |
| ≦13 years | 601 | (33.4) | 186 | (26.3) | 1,157 | (21.7) | 254 | (36.7) | 2,198 | (25.7) |
| others | 4 | (0.2) | 4 | (0.6) | 32 | (0.6) | 3 | (0.4) | 43 | (0.5) |
| Missing | | | | | | | | | 42 | |
| **Annual household income, n(%)** | | | | | | | | | | |
| ≦1.49 million yen | 432 | (25.9) | 255 | (37.9) | 1,923 | (41.2) | 231 | (36.3) | 2,841 | (37.2) |
| 1.50–2.99 million yen | 867 | (52.0) | 325 | (48.3) | 2,245 | (48.1) | 303 | (47.6) | 3,740 | (48.9) |
| 3.00–5.99 million yen | 295 | (17.7) | 83 | (12.3) | 409 | (8.8) | 88 | (13.8) | 875 | (11.4) |
| ≧6.00 million yen | 74 | (4.4) | 10 | (1.5) | 91 | (1.9) | 15 | (2.4) | 190 | (2.5) |
| Missing | | | | | | | | | 933 | |

Table 5 shows the logistic regression analysis of SWB in 2019 on interpersonal relationships and independent functional factors. Sex and SWB levels at baseline were adjustment variables. Among married men, all independent functional factors were associated with SWB except for the frequency of eating meat and fish, alcohol consumption, and instrumental self-maintenance. In addition, except for involvement in sports groups or clubs, health promotion activities, and intergenerational exchange, all interpersonal relationship factors were associated with SWB. Among unmarried men, independent functional factors such as frequency of eating meat and fish (OR = 1.52, 95% Cl: 1.13–2.05), smoking (OR = 0.69, 95% Cl: 0.53–0.90), self-reported health (OR = 1.56, 95% Cl: 1.08–2.26), and GDS (OR = 0.84, 95% Cl: 0.78–0.91) were associated with SWB. In addition, interpersonal relationship factors such as receiving emotional support (OR = 1.34, 95% Cl: 1.05–1.71) and health promotion activities (OR = 3.06, 95% Cl: 1.37–6.84) were associated with SWB. Among married women, all independent functional factors were associated with SWB except for alcohol consumption. In addition, except for membership in senior citizens' clubs, all interpersonal relationship factors were associated with SWB. Among unmarried women, independent functional factors such as self-reported health (OR = 1.83, 95% Cl: 1.23–2.72) and GDS (OR = 0.82, 95% Cl: 0.75–0.90) were associated with SWB. In addition, interpersonal relationship factors such as the receipt and provision of social support, the number of types of social participation, and involvement in volunteer groups, neighborhood associations, and intergenerational exchange were associated with SWB.

## Discussion

This is the first study to examine the factors associated with SWB according to sex and marital status among community-dwelling older adults living alone in Japan. We used data from the JAGES large-scale database project of people over 65 years of age. Participants in the JAGES were recruited from municipalities throughout Japan and were deemed not to require long-term care by a "long-term care need certification system" in Japan. Therefore, the study's target population can be regarded as representative of older community-dwelling adults living alone in Japan.

**Table 3. Predictors of study participants by sex and marital status.**

| Predictors | Married Men | | Unmarried Men | | Married Women | | Unmarried Women | | All | |
|---|---|---|---|---|---|---|---|---|---|---|
| | n = | 1,803 | n = | 709 | n = | 5,372 | n = | 695 | n = | 8,579 |
| **Independent functional factors** | | | | | | | | | | |
| **Frequency of eating meat and fish, n(%)** | | | | | | | | | | |
| Less than once a week | 192 | (10.8) | 94 | (13.4) | 280 | (5.3) | 36 | (5.3) | 602 | (7.1) |
| 2–6 times per week | 899 | (50.6) | 361 | (51.6) | 2,340 | (44.2) | 313 | (45.8) | 3,913 | (46.3) |
| everyday | 687 | (38.6) | 244 | (34.9) | 2,675 | (50.5) | 335 | (49.0) | 3,941 | (46.6) |
| Missing | | | | | | | | | 123 | |
| **Frequency of eating vegetables and fruits, n(%)** | | | | | | | | | | |
| Less than once a week | 128 | (7.2) | 61 | (8.7) | 62 | (1.2) | 8 | (1.2) | 259 | (3.0) |
| 2–6 times per week | 612 | (34.4) | 272 | (38.7) | 745 | (14.0) | 78 | (11.3) | 1,707 | (20.1) |
| everyday | 1,039 | (58.4) | 370 | (52.6) | 4,514 | (84.8) | 604 | (87.5) | 6,527 | (76.9) |
| Missing | | | | | | | | | 86 | |
| **Alcohol consumption, n(%)** | | | | | | | | | | |
| Never | 410 | (23.1) | 199 | (28.5) | 3,494 | (67.4) | 400 | (59.0) | 4,503 | (54.0) |
| Past | 295 | (16.6) | 114 | (16.3) | 463 | (8.9) | 83 | (12.2) | 955 | (11.5) |
| Currently | 1,069 | (60.3) | 386 | (55.2) | 1,229 | (23.7) | 195 | (28.8) | 2,879 | (34.5) |
| Missing | | | | | | | | | 242 | |
| **Smoking status, n(%)** | | | | | | | | | | |
| Never | 436 | (24.5) | 199 | (28.3) | 4,527 | (85.8) | 560 | (81.4) | 5,722 | (67.7) |
| Past | 891 | (50.1) | 312 | (44.4) | 451 | (8.5) | 95 | (13.8) | 1,749 | (20.7) |
| Currently | 453 | (25.4) | 192 | (27.3) | 301 | (5.7) | 33 | (4.8) | 979 | (11.6) |
| Missing | | | | | | | | | 129 | |
| **Health** | | | | | | | | | | |
| **Self-reported helth, n(%)** | | | | | | | | | | |
| Poor | 22 | (1.2) | 11 | (1.6) | 39 | (0.7) | 6 | (0.9) | 78 | (0.9) |
| Fair | 236 | (13.4) | 122 | (17.5) | 492 | (9.4) | 59 | (8.7) | 909 | (10.8) |
| good | 1,268 | (71.8) | 490 | (70.4) | 3,826 | (73.0) | 507 | (74.6) | 6,091 | (72.6) |
| very good | 240 | (13.6) | 73 | (10.5) | 887 | (16.9) | 108 | (15.9) | 1,308 | (15.6) |
| Missing | | | | | | | | | 193 | |
| **TMIG-IC, mean±SD(score)** | | | | | | | | | | |
| TIMG total score | 11.28 | ±1.61 | 10.58 | ±1.78 | 11.97 | ±1.29 | 11.85 | ±1.39 | 11.70 | ±1.48 |
| Instrumental Self-Maintenance | 4.98 | ±0.14 | 4.97 | ±0.23 | 4.98 | ±0.18 | 4.98 | ±0.22 | 4.98 | ±0.18 |
| Intellectual Activity | 3.49 | ±0.78 | 3.37 | ±0.83 | 3.60 | ±0.68 | 3.63 | ±0.61 | 3.56 | ±0.71 |
| Social Role | 2.80 | ±1.19 | 2.24 | ±1.30 | 3.37 | ±0.92 | 3.22 | ±1.01 | 3.14 | ±1.08 |
| **GDS score, mean±SD(score)** | 3.89 | ±3.70 | 4.45 | ±3.75 | 2.89 | ±2.96 | 2.94 | ±3.18 | 3.25 | ±3.27 |
| **Interpersonal relationships factors** | | | | | | | | | | |
| **Frequency of contact with friends, n(%)** | | | | | | | | | | |
| No contact | 181 | (10.2) | 144 | (20.7) | 181 | (3.4) | 29 | (4.2) | 535 | (6.4) |
| A few times a year | 328 | (18.4) | 142 | (20.4) | 552 | (10.5) | 108 | (15.7) | 1,130 | (13.4) |
| 1–3 times a month | 389 | (21.8) | 154 | (22.1) | 1,128 | (21.5) | 183 | (26.7) | 1,854 | (22.0) |
| Once a week | 233 | (13.1) | 92 | (13.2) | 763 | (14.5) | 78 | (11.4) | 1,166 | (13.8) |
| 2–3 times a week | 316 | (17.7) | 93 | (13.4) | 1,392 | (26.5) | 177 | (25.8) | 1,978 | (23.5) |
| More than 4 times a week | 334 | (18.8) | 71 | (10.2) | 1,240 | (23.6) | 111 | (16.2) | 1,756 | (20.9) |
| Missing | | | | | | | | | 160 | |
| **Relationship with neighbors, n(%)** | | | | | | | | | | |
| No socializing with neighbors | 143 | (8.0) | 92 | (13.2) | 110 | (2.1) | 21 | (3.0) | 366 | (4.3) |
| No more than exchanging greetings | 715 | (40.1) | 339 | (48.8) | 1,029 | (19.4) | 221 | (31.9) | 2,304 | (27.2) |

*(Continued)*

**Table 3.** (Continued)

| Predictors | Married Men | | Unmarried Men | | Married Women | | Unmarried Women | | All | |
|---|---|---|---|---|---|---|---|---|---|---|
| | n = | 1,803 | n = | 709 | n = | 5,372 | n = | 695 | n = | 8,579 |
| Chatting with neighbors | 793 | (44.5) | 227 | (32.7) | 2,946 | (55.5) | 351 | (50.7) | 4,317 | (50.9) |
| Cooperation in daily life | 132 | (7.4) | 37 | (5.3) | 1,219 | (23.0) | 99 | (14.3) | 1,487 | (17.5) |
| Missing | | | | | | | | | 105 | |
| **Social support** | | | | | | | | | | |
| **Social support numbers, mean±SD (number of person)** | | | | | | | | | | |
| Receiving emotional support | 1.22 | ±0.95 | 0.86 | ±0.78 | 2.01 | ±1.01 | 1.56 | ±0.80 | 1.71 | ±1.05 |
| Providing emotional support | 1.26 | ±0.97 | 0.90 | ±0.83 | 1.98 | ±1.03 | 1.62 | ±0.83 | 1.71 | ±1.05 |
| Receiving instrumental support | 0.87 | ±0.76 | 0.54 | ±0.64 | 1.27 | ±0.82 | 0.99 | ±0.75 | 1.10 | ±0.83 |
| Providing instrumental support | 0.63 | ±0.79 | 0.42 | ±0.60 | 1.17 | ±0.97 | 0.97 | ±0.78 | 0.98 | ±0.94 |
| **Social participation** | | | | | | | | | | |
| Number of social participations, mean±SD | 1.61 | ±1.93 | 1.01 | ±1.42 | 2.25 | ±2.11 | 2.01 | ±1.91 | 1.97 | ±2.03 |
| **Volunteer groups, n(%)** | | | | | | | | | | |
| Non-participation | 1,279 | (78.6) | 578 | (88.0) | 3,296 | (71.7) | 498 | (77.9) | 5,651 | (75.1) |
| Participation | 349 | (21.4) | 79 | (12.0) | 1,300 | (28.3) | 141 | (22.1) | 1,869 | (24.9) |
| Missing | | | | | | | | | 1,059 | |
| **Sports groups or clubs, n(%)** | | | | | | | | | | |
| Non-participation | 1,064 | (67.4) | 515 | (81.9) | 2,501 | (56.7) | 377 | (60.1) | 4,457 | (61.5) |
| Participation | 514 | (32.6) | 114 | (18.1) | 1,907 | (43.3) | 250 | (39.9) | 2,785 | (38.5) |
| Missing | | | | | | | | | 1,337 | |
| **Hobby groups, n(%)** | | | | | | | | | | |
| Non-participation | 982 | (59.6) | 478 | (74.3) | 2,093 | (44.0) | 310 | (47.8) | 3,863 | (50.2) |
| Participation | 665 | (40.4) | 165 | (25.7) | 2,660 | (56.0) | 338 | (52.2) | 3,828 | (49.8) |
| Missing | | | | | | | | | 888 | |
| **Senior citizen clubs, n(%)** | | | | | | | | | | |
| Non-participation | 1,382 | (84.0) | 610 | (95.2) | 3,666 | (78.6) | 589 | (91.7) | 6,247 | (82.3) |
| Participation | 264 | (16.0) | 31 | (4.8) | 997 | (21.4) | 53 | (8.3) | 1,345 | (17.7) |
| Missing | | | | | | | | | 987 | |
| **Neighborhood associations, n(%)** | | | | | | | | | | |
| Non-participation | 1,075 | (65.0) | 470 | (73.0) | 2,767 | (59.2) | 442 | (68.7) | 4,754 | (62.4) |
| Participation | 580 | (35.0) | 174 | (27.0) | 1,910 | (40.8) | 201 | (31.3) | 2,865 | (37.6) |
| Missing | | | | | | | | | 960 | |
| **Learning or Culture groups, n(%)** | | | | | | | | | | |
| Non-participation | 1,448 | (88.7) | 608 | (94.7) | 3,516 | (75.8) | 452 | (71.0) | 6,024 | (79.8) |
| Participation | 185 | (11.3) | 34 | (5.3) | 1,125 | (24.2) | 185 | (29.0) | 1,529 | (20.2) |
| Missing | | | | | | | | | 1,026 | |
| **Health promotion activities, n(%)** | | | | | | | | | | |
| Non-participation | 1,470 | (89.6) | 606 | (94.0) | 3,607 | (76.9) | 533 | (82.8) | 6,216 | (81.6) |
| Participation | 170 | (10.4) | 39 | (6.0) | 1,085 | (23.1) | 111 | (17.2) | 1,405 | (18.4) |
| Missing | | | | | | | | | 958 | |
| **Intergenerational exchange, n(%)** | | | | | | | | | | |
| Non-participation | 1,451 | (87.1) | 605 | (93.1) | 4,059 | (85.5) | 550 | (84.7) | 6,665 | (86.4) |
| Participation | 214 | (12.9) | 45 | (6.9) | 690 | (14.5) | 99 | (15.3) | 1,048 | (13.6) |
| Missing | | | | | | | | | 866 | |

TMIG-IC: Tokyo Metropolitan Institute of Gerontology Index of Competence.

**Table 4. Transitions of SWB from 2016 to 2019 according to sex and marital status.**

| Transitions of SWB[a] | Married Men | | Unmarried Men | | Married Women | | Unmarried Women | | All | |
|---|---|---|---|---|---|---|---|---|---|---|
| | n = 1803, n(%) | | n = 709, n(%) | | n = 5372, n(%) | | n = 695, n(%) | | n = 8579, n(%) | |
| High to High | 588 | (32.6) | 150 | (21.2) | 2,014 | (37.5) | 220 | (31.7) | 2,972 | (34.6) |
| Low to High | 218 | (12.1) | 70 | (9.9) | 636 | (11.8) | 66 | (9.5) | 990 | (11.5) |
| High to Low | 189 | (10.5) | 66 | (9.3) | 667 | (12.4) | 73 | (10.5) | 995 | (11.6) |
| Low to Low | 808 | (44.8) | 423 | (59.7) | 2,055 | (38.3) | 336 | (48.3) | 3,622 | (42.2) |

[a] Based on the median SWB by gender, we defined High > 6 for Men and High > 7 for Women.

Among married men, several independent functional and interpersonal relationship factors effectively drove high-level SWB. Those who consumed vegetables and fruits more frequently and did not smoke maintained a higher SWB. Older adults living alone tended to consume less meat, fish, vegetables, and fruits; especially older men living alone consumed fewer vegetables and fruits [27]. Prior research has shown that increased consumption of fruits and vegetables predicts improved well-being [28]. A healthy lifestyle is important in healthy aging and is associated with outcomes such as positive health status and an affirmative self-perceived meaning of life in older adults [29]. Among interpersonal relationship factors, more opportunities for social relations, social support, and social participation contributed to high-level SWB in married men. Older adults who live alone and have relatives and non-relatives in their social network are more satisfied with life than older adults who live with others [30]. Although living alone is often considered a risk for older adults, this study suggests a strong probability that these adults can progress to high-level SWB by being involved with others.

The fewest number of factors significantly driving SWB were observed among unmarried men compared with the remaining categories. Among interpersonal relationship factors, only receipt of emotional support and participation in health promotion activities contributed to high-level SWB in unmarried men. In this study, unmarried men were the least likely to receive and provide support and participate in social activities (Table 3). Men and never-married older adults had fewer possibilities of contact with relatives and friends and their social networks tended to be more restricted [30]. Because these groups prefer to interact less with others and have few interpersonal relationships, they do not receive the benefits that are derived from interpersonal factors. Unlike members in other categories, these groups were not significantly affected by activities involving others, such as providing support and social participation; however, receiving support was one of the factors associated with high-level SWB in these groups. Previous research indicates that receiving support from friends and family influences SWB in men [31]. Because unmarried men have few interpersonal relationships, an effective intervention for improving SWB in this group may be to increase the quality and quantity of appropriate social support provided.

Personal and interpersonal factors were important for high-level SWB in married women. The greatest number of significant factors was observed in this group; all independent functional factors except alcohol consumption were associated with SWB. As with married men, a healthy lifestyle may contribute to SWB in this group. Although the association was not observed in men and unmarried women, higher IADL functioning factors were associated with higher SWB in married women—similar to married men. Low Activity of Daily Living (ADL) is known to be associated with low life satisfaction [32]. This study further supported that IADL—a higher level of life function than ADL—was associated with SWB. These results emphasize the importance of living independently. Members of the unmarried groups have

**Table 5. Logistic regression analysis of SWB in 2019 on predictors.**

| Predictors | Married Men | | Unmarried Men | | Married Women | | Unmarried Women | |
|---|---|---|---|---|---|---|---|---|
| | OR | 95%Cl | OR | 95%Cl | OR | 95%Cl | OR | 95%Cl |
| Age [a] | 1.03*** | (1.02, 1.05) | 1.00 | (0.96, 1.04) | 1.03*** | (1.02, 1.04) | 1.02 | (0.99, 1.04) |
| SWB H and L group at 2016 [a] | 11.53*** | (9.24, 14.40) | 13.73*** | (9.35, 20.17) | 9.76*** | (8.61, 11.05) | 15.34*** | (10.56, 22.30) |
| **Independent functional factors[b]** | | | | | | | | |
| **Lifestyle** | | | | | | | | |
| Frequency of eating meat and fish | 1.18 | (0.99, 1.40) | 1.52** | (1.13, 2.05) | 1.18** | (1.06, 1.32) | 1.07 | (0.78, 1.47) |
| Frequency of eating vegetables and fruits | 1.49*** | (1.23, 1.80) | 1.20 | (0.88, 1.62) | 1.32*** | (1.12, 1.55) | 1.60 | (0.97, 2.65) |
| Alcohol consumption | 1.06 | (0.93, 1.22) | 0.95 | (0.76, 1.18) | 1.02 | (0.94, 1.10) | 0.90 | (0.72, 1.11) |
| Smoke | 0.72*** | (0.61, 0.84) | 0.69** | (0.53, 0.90) | 0.80*** | (0.70, 0.90) | 0.90 | (0.62, 1.31) |
| **Health** | | | | | | | | |
| Self-reported health | 1.97*** | (1.57, 2.46) | 1.56* | (1.08, 2.26) | 1.79*** | (1.58, 2.04) | 1.83** | (1.23, 2.72) |
| **TMIG-IC** | | | | | | | | |
| TIMG score | 1.28*** | (1.18, 1.38) | 1.06 | (0.95, 1.19) | 1.23*** | (1.16, 1.29) | 1.14 | (0.98, 1.33) |
| Instrumental Self-Maintenance | 2.00 | (0.86, 4.63) | 1.49 | (0.44, 5.07) | 1.66* | (1.12, 2.45) | 1.40 | (0.33, 5.87) |
| Intellectual Activity | 1.29** | (1.11, 1.51) | 1.04 | (0.82, 1.33) | 1.20*** | (1.09, 1.32) | 1.39 | (0.99, 1.93) |
| Social Role | 1.36*** | (1.23, 1.50) | 1.09 | (0.93, 1.27) | 1.32*** | (1.22, 1.42) | 1.13 | (0.93, 1.38) |
| **GDS** | | | | | | | | |
| GDS score | 0.80*** | (0.77, 0.84) | 0.84*** | (0.78, 0.91) | 0.80*** | (0.78, 0.83) | 0.82*** | (0.75, 0.90) |
| **Interpersonal relationships factors[b]** | | | | | | | | |
| Frequency of contact with friends | 1.09* | (1.02, 1.17) | 1.00 | (0.89, 1.13) | 1.11*** | (1.06, 1.16) | 1.07 | (0.94, 1.22) |
| relationship with neighbors | 1.45*** | (1.24, 1.69) | 1.08 | (0.83, 1.40) | 1.23*** | (1.13, 1.35) | 1.03 | (0.79, 1.34) |
| **Social support** | | | | | | (,) | | |
| Receiving emotional support | 1.17** | (1.04, 1.32) | 1.34* | (1.05, 1.71) | 1.22*** | (1.15, 1.30) | 1.42** | (1.12, 1.80) |
| Providing emotional support | 1.24*** | (1.10, 1.40) | 1.16 | (0.92, 1.47) | 1.22*** | (1.15, 1.30) | 1.46** | (1.15, 1.84) |
| Receiving instrumental support | 1.44*** | (1.24, 1.69) | 1.23 | (0.91, 1.66) | 1.28*** | (1.18, 1.38) | 1.44** | (1.12, 1.87) |
| Providing instrumental support | 1.27** | (1.10, 1.47) | 1.32 | (0.95, 1.82) | 1.25*** | (1.16, 1.33) | 1.39** | (1.09, 1.78) |
| **Social participation** | | | | | | | | |
| Number of social participations | 1.10** | (1.03, 1.18) | 1.08 | (0.94, 1.24) | 1.12*** | (1.08, 1.16) | 1.12* | (1.00, 1.24) |
| Volunteer group | 1.41* | (1.05, 1.87) | 1.02 | (0.57, 1.84) | 1.42*** | (1.22, 1.65) | 1.91** | (1.20, 3.05) |
| Sports groups or clubs | 1.17 | (0.90, 1.51) | 1.86 | (1.12, 3.07) | 1.42*** | (1.23, 1.63) | 1.08 | (0.72, 1.61) |
| Hobby Groups | 1.48** | (1.16, 1.88) | 1.25 | (0.79, 1.95) | 1.27*** | (1.11, 1.45) | 1.18 | (0.80, 1.74) |
| Club of the elderly | 1.42* | (1.02, 1.96) | 1.41 | (0.57, 3.48) | 1.13 | (0.96, 1.34) | 1.04 | (0.51, 2.11) |
| Neighborhood association | 1.43** | (1.12, 1.82) | 0.90 | (0.57, 1.41) | 1.33*** | (1.16, 1.52) | 1.52* | (1.00, 2.29) |
| Learning and Culture circle | 1.90*** | (1.30, 2.79) | 1.03 | (0.44, 2.40) | 1.43*** | (1.22, 1.68) | 1.19 | (0.78, 1.81) |
| Health promotion activities | 1.29 | (0.88, 1.90) | 3.06** | (1.37, 6.84) | 1.31*** | (1.12, 1.54) | 1.29 | (0.78, 2.15) |
| Activities to share your skills and experiences with others | 1.32 | (0.93, 1.87) | 1.41 | (0.66, 3.00) | 1.55*** | (1.28, 1.87) | 2.06** | (1.21, 3.50) |

[a] Univariate analysis.

[b] Adjustment in 2016 for age and SWB in the H and L groups, respectively.

*** p<0.001

**p<0.01

*p<0.05.

managed their lives by themselves for long periods. In contrast, those in the married groups are considered to have lived with someone for a long time; we infer that they experienced a change in their lives when they became single. The ability to overcome change and live independently may drive high SWB. In addition, all interpersonal relationship factors except for membership in senior citizens' clubs were associated with SWB. Previous studies showed that

the number of types of social participation contributes to a positive effect on SWB [33,34]. This may stem from the fact that most of the older adults in the previous study population were married and women.

Although interpersonal factors were significant in many cases among unmarried women, fewer factors predicted high-level SWB in this group compared with married women. Among unmarried women in this study, activities such as volunteering and involvement in neighborhood associations and intergenerational exchanges were effective factors. Engaging in activities—such as volunteering—that assign specific functional roles rather than activities that only enhance enjoyment and pleasure may be effective in improving well-being outcomes [35,36]. For never-married persons, who are expected to live alone for longer and have fewer opportunities to take on roles through their children, family members, and relatives than married persons, the results suggest that taking on roles is fundamentally important for unmarried women. In this study, providing support was significantly associated with SWB in both men and women, except in unmarried men. This factor cannot be ignored as a means for maintaining high-level SWB, especially in unmarried women. Previous studies suggest that providing support contributes to SWB more than receiving support; providing support to children and friends is especially effective [37]. Creating opportunities for unmarried women to assume roles and provide support may be critical for driving SWB.

The originality of this study derives from classifying married, divorced, and widowed persons as "married" and never-married persons as "unmarried." As a result, we observed that the impact of SWB factors varied by sex and marital status among older people living alone. These findings are useful for policy-making and intervention activities to prevent health problems in a society in which the marital status of older adults is changing dramatically.

## Limitations

Our study had several limitations. First, although this was a longitudinal study that shows how status at baseline predicts SWB years later, we adjusted for only age and baseline SWB level; therefore, we may not have eliminated confounding factors. Second, the number of participants in the unmarried groups was smaller than in the married groups, which may have resulted in a difference in the power of detection. Nevertheless, despite the small percentage of never-married persons in the total population in Japan, our ability to secure a large and representative number of never-married persons living alone throughout Japan is a study strength.

## Author Contributions

**Conceptualization:** Nana Abe, Nanami Oe, Etsuko Tadaka.

**Data curation:** Nana Abe.

**Formal analysis:** Nana Abe.

**Funding acquisition:** Etsuko Tadaka.

**Investigation:** Nana Abe.

**Methodology:** Nana Abe, Nanami Oe, Etsuko Tadaka, Toshiyuki Ojima.

**Project administration:** Etsuko Tadaka.

**Resources:** Etsuko Tadaka.

**Software:** Nana Abe, Nanami Oe.

**Supervision:** Toshiyuki Ojima.

**Validation:** Nana Abe, Nanami Oe, Etsuko Tadaka.

**Visualization:** Nana Abe, Nanami Oe.

**Writing – original draft:** Nana Abe, Nanami Oe.

**Writing – review & editing:** Nana Abe, Nanami Oe, Etsuko Tadaka, Toshiyuki Ojima.

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
