## [Decision Letter · Decision Letter 0]

20 Apr 2023

PONE-D-23-04435Factors Related to Subjective Well-being Among Community-Dwelling Older Adults Living Alone: A Stratified Analysis by Sex and Marital Status from the JAGESPLOS ONE

Dear Dr. Tadaka,

Thank you for submitting your manuscript to PLOS ONE. After careful consideration, we feel that it has merit but does not fully meet PLOS ONE’s publication criteria as it currently stands. Therefore, we invite you to submit a revised version of the manuscript that addresses the points raised during the review process.

It is an interesting study in which authors investigated actors related to subjective well-being among community-dwelling Older Adults Living Alone, which is kind of significant implicaition. However, some technical issues, as mentioned by the reviewers, should be addressed in order to further improve the mauscript in terms of scientificity, especially about methodology.

We look forward to receiving your revised manuscript.

Kind regards,

Shaonong Dang, PhD

Academic Editor

PLOS ONE

Reviewers' comments:

Reviewer's Responses to Questions

**Comments to the Author**

1. Is the manuscript technically sound, and do the data support the conclusions?

Reviewer #1: No

Reviewer #2: Yes

Reviewer #3: Yes

2. Has the statistical analysis been performed appropriately and rigorously? 

Reviewer #1: No

Reviewer #2: Yes

Reviewer #3: Yes

3. Have the authors made all data underlying the findings in their manuscript fully available?

Reviewer #1: Yes

Reviewer #2: Yes

Reviewer #3: Yes

4. Is the manuscript presented in an intelligible fashion and written in standard English?

Reviewer #1: Yes

Reviewer #2: Yes

Reviewer #3: Yes

5. Review Comments to the Author

Reviewer #1: The comments are as follows:

1. I am really confused about the objective and hypothesis of the present study.

2. I think authors have done the polled data analysis...If yes? why they did not adopted other longitudinal data analysis techniques?

3. The data is so rich; therefore using a pooled data analysis which is largely appropriate for cross-sectional studies is a matter of concern for me.

Reviewer #2: Comments

This study examined factors related to subjective well-being (SWB) among Japanese community-dwelling elderly stratified by sex and marital status. This manuscript may contribute to this area of research. Further attention to the issues presented below would strengthen the manuscript.

Major points

#1

Regarding participant groups.

In this study, only older adults who live alone were included in the analysis, and they were divided into groups by gender and marital status to compare their SWB.

It would be better to state in the text body about the reason why the analysis was conducted only for those living alone in this study. If the authors are interested in the SWB states of older adults who live alone, it would be better to have a comparison group (i.e., older individuals who live with their families) to better clarify the nature of older adults living alone.

For those who are married but living alone, it would be helpful to specify their specific situation in the method (e.g., married, but their husband institutionalized?). How many of these are included in the data set? Are there any problems in analyzing these as the same group as those who have divorced from spouses or have been bereaved of his/her spouse?

#2

Regarding outcomes.

The authors should specify in the statistical analyses section how the outcomes (binary values) of the longitudinal analyses were defined and analyzed. Also, since it appears that the baseline values were controlled for in the multivariate analysis (Table 5), the statistical adjustments should be clearly stated in the text. In addition, the proportion of events occurring in the outcome should be noted in the results.

#3

Regarding multivariate analysis.

It should be clearly stated whether all the independent variables were entered simultaneously or not in multivariate analysis. In particular, interpersonal relationship factors include variables that have a high correlation with each other, but are there any problems in such analysis? Is there any possibility of over-adjustment?

Similarly, if the subscales of the TMIG-IC were simultaneously entered into the model, would this pose any problems? It is theoretically unlikely that the total score and the subscale score would be entered at the same time, I believe. The authors' opinion should be stated on this point.

Minor points

#4

In Table 5, TMIG-IC subscale names should be used from the official names provided by the previous study (Koyano et al. 1991).

#5

Typos?

Notes in Table5. SWB H and group  SWB H and L group

Reviewer #3: 1. In the Abstract, what is the meaning of "SWB" ? Pleased add the information at this word firstly appeared.

2. In the table 1, add the collected time of variables.

3. “How happy are you currently? Please provide a score on a 10-point scale from 0 (very unhappy) to 10 (very happy).” Is this question suitable for assessing the SWB? Added reference.

4. How to assess the the transitions of marital status from 2016 ot 2019? And presented the associaiton between the transition of SWB and transitions of marital status?

5.Why used logistic regression inseaded of COX?

6. PLOS authors have the option to publish the peer review history of their article (what does this mean?). If published, this will include your full peer review and any attached files.

Reviewer #1: No

Reviewer #2: No

Reviewer #3: No

---

## [Author Response · Author response to Decision Letter 0]

26 Jun 2023

To Reviewer #1

Thank you for the thoughtful and constructive feedback you provided regarding our manuscript, Factors Related to Subjective Well-being Among Community-Dwelling Older Adults Living Alone: A Stratified Analysis by Sex and Marital Status from the JAGES (PONE-D-23-04435).

1. I am really confused about the objective and hypothesis of the present study.

⇒The number of older adults living alone, which is expected to increase in Japan and other developed countries, is likely to be diverse with regard to sex and marital status. Therefore, focusing on the four population categories of older adults living alone based on sex and marital status (married men, unmarried men, married women and unmarried women) —rather than on the overall group of older adults living alone—while also focusing on not only negative health outcomes but also positive health outcomes such as SWB is necessary. Moreover, studying representative populations in longitudinal studies from which causal relationships between SWB over time and their association can be inferred is critical for developing evidence-based health policy that is responsive to new populations and diversity. Therefore, in this study, by using a longitudinal study of a representative population (the Japan Agency for Gerontological Evaluation Study), our aim was to determine changes in SWB over time and their association with independent functioning factors and interpersonal factors by sex and marital status (married men, unmarried men, married women and unmarried women) among older adults living alone in the community. Independent functioning factors are critical for performing daily living tasks without help [21, 22]. The achievement of functional independence ensures that older adults can participate fully in meaningful and purposeful life events—a requirement for SWB. Interpersonal factors are individual elements or a group of factors that positively or negatively influence the quality of relationships [21, 23] and are integral to creating and maintaining meaningful personal relationships that ensure SWB among older adults in the community. Our hypotheses were as follows:

Hypothesis 1: SWB is more likely to transition to a high level in women rather than men and in married versus unmarried individuals among community-dwelling older adults living alone. 

Hypothesis 2: Independent functioning factors and interpersonal factors are significantly associated with SWB in older adults living alone and differ by sex and marital status among community-dwelling older adults living alone. (L95～)

2. I think authors have done the polled data analysis...If yes? why they did not adopted other longitudinal data analysis techniques?

3. The data is so rich; therefore using a pooled data analysis which is largely appropriate for cross-sectional studies is a matter of concern for me.

⇒This study is not a polled data analysis but a longitudinal data analysis (⇒the statistical method). 

 

To Reviewer #2

Thank you for the thoughtful and constructive feedback you provided regarding our manuscript, Factors Related to Subjective Well-being Among Community-Dwelling Older Adults Living Alone: A Stratified Analysis by Sex and Marital Status from the JAGES (PONE-D-23-04435).

This study examined factors related to subjective well-being (SWB) among Japanese community-dwelling elderly stratified by sex and marital status. This manuscript may contribute to this area of research. Further attention to the issues presented below would strengthen the manuscript.

Major points

#1Regarding participant groups.

In this study, only older adults who live alone were included in the analysis, and they were divided into groups by gender and marital status to compare their SWB.

a) It would be better to state in the text body about the reason why the analysis was conducted only for those living alone in this study. If the authors are interested in the SWB states of older adults who live alone, it would be better to have a comparison group (i.e., older individuals who live with their families) to better clarify the nature of older adults living alone. b)For those who are married but living alone, it would be helpful to specify their specific situation in the method (e.g., married, but their husband institutionalized?). How many of these are included in the data set? Are there any problems in analyzing these as the same group as those who have divorced from spouses or have been bereaved of his/her spouse?

⇒Thank you for your comments. 　a) We agree with you that it would be better to have a comparison group (i.e., older individuals who live with their families) to better clarify the nature of older adults living alone. However, while it is already clear that older adults living alone are risk factors for a various health outcome, including SWB, compared to older adults living with family members, it is not clear whether SWB and related factors differ by sex and marital status among older adults living alone. Therefore, in this study, we analyzed a sample of older adults living alone, stratified by sex and marital status (married men, non-married men, married women, and non-married women).　b) Those who are married but living alone situation assumes that the spouse is an institutionalized resident, as you pointed out. Since the main purpose of this survey is to compare men and women, married and unmarried, those who were divorced, bereaved, or separated (institutionalized) were included in the married group. We would like to research on comparison of divorced, bereaved, or separated (institutionalized) in the married group as you recommended in the next step. Thank you for the thoughtful feedback.

Additional information: The married men group consisted of 1,803 men, among whom 63 (3.5%) were currently married, 1,005 (55.7%) were widowed, and 735 (40.8%) were divorced. The married women group consisted of 1,803 women, among whom 63 (3.5%) were married, 1005 (55.7%) were widowed, and 735 (40.8%) were divorced. In addition, individuals in the “married with a spouse” group were assumed to be those whose spouses lived alone because they were moved to an institution or for other reasons.　(L.152～)

 

#2

Regarding outcomes.

The authors should specify in the statistical analyses section how the outcomes (binary values) of the longitudinal analyses were defined and analyzed. Also, since it appears that the baseline values were controlled for in the multivariate analysis (Table 5), the statistical adjustments should be clearly stated in the text. In addition, the proportion of events occurring in the outcome should be noted in the results.

⇒Thank you for your important comments. We have added a note about a detailed description of the analytical method in statistical analyses section, as follows: The logistic regression analysis was conducted with the dependent variables as “1” for high level SWB in 2019 and “0” for low level SWB in 2019. The analysis was conducted with sex and SWB level at baseline as adjustment variable (L.229～). 

This study did not evaluate the occurrence of SWB-related events (e.g., death of a friend). This is the next issue as you commented.

#3

Regarding multivariate analysis.

It should be clearly stated whether all the independent variables were entered simultaneously or not in multivariate analysis. In particular, interpersonal relationship factors include variables that have a high correlation with each other, but are there any problems in such analysis? Is there any possibility of over-adjustment?

Similarly, if the subscales of the TMIG-IC were simultaneously entered into the model, would this pose any problems? It is theoretically unlikely that the total score and the subscale score would be entered at the same time, I believe. The authors' opinion should be stated on this point.

⇒As you indicated, we did not perform a multivariate analysis because of multicollinearity. Univariate logistic regression analyses were performed for each independent variable, and in each univariate logistic regression analysis, gender and SWB level at baseline were entered and adjusted. Similarly, univariate logistic regression analysis was also performed for the total score and subscales, taking into account multicollinearity for the TMIG-IC.

Minor points

#4

In Table 5, TMIG-IC subscale names should be used from the official names provided by the previous study (Koyano et al. 1991).

⇒We are sorry. The Tokyo Metropolitan Institute of Gerontology Index of Competence (TMIG-IC). [22]. (L.181～)

#5

Typos?

Notes in Table5. SWB H and group  SWB H and L group

⇒We are sorry. Notes in Table5 has been revised.

 

To Reviewer #3

Thank you for the thoughtful and constructive feedback you provided regarding our manuscript, Factors Related to Subjective Well-being Among Community-Dwelling Older Adults Living Alone: A Stratified Analysis by Sex and Marital Status from the JAGES (PONE-D-23-04435).

1. In the Abstract, what is the meaning of "SWB" ? Pleased add the information at this word firstly appeared.

⇒We are sorry and thank you for your comment. We added the information of subjective well-being (SWB) in the Abstract.

2. In the table 1, add the collected time of variables.

⇒Thank you for your suggestion. We added the collected time of variables in table 1.

3. “How happy are you currently? Please provide a score on a 10-point scale from 0 (very unhappy) to 10 (very happy).” Is this question suitable for assessing the SWB? Added reference.

⇒Thank you for your suggestion. The SWB 10-point scale has been used in JAGES. We have added a reference regarding SWB in JAGES [21].

4. How to assess the transitions of marital status from 2016 to 2019? And presented the association between the transition of SWB and transitions of marital status?

⇒Thank you for your comment. We did not assessed the transitions in marital status from 2016 to 2019 since we exclude subject whose marital status has changed. 

5.Why used logistic regression inseaded of COX?

⇒Thank you for your comment. We used logistic regression since Cox proportional hazards regression assumes proportionality, but logistic regression is not subject to this restriction.

---

## [Decision Letter · Decision Letter 1]

5 Jul 2023

PONE-D-23-04435R1Factors Related to Subjective Well-being Among Community-Dwelling Older Adults Living Alone: A Stratified Analysis by Sex and Marital Status from the JAGESPLOS ONE

Dear Dr. Tadaka,

Thank you for submitting your manuscript to PLOS ONE. After careful consideration, we feel that it has merit but does not fully meet PLOS ONE’s publication criteria as it currently stands. Therefore, we invite you to submit a revised version of the manuscript that addresses the points raised during the review process.

Authors have addressed most of comments from the reviewers. Now a few minor concerns have been raised, so authors are suggested to address further them in order to improve the manuscript.

We look forward to receiving your revised manuscript.

Kind regards,

Shaonong Dang, PhD

Academic Editor

PLOS ONE

Journal Requirements:

Reviewers' comments:

Reviewer's Responses to Questions

**Comments to the Author**

1. If the authors have adequately addressed your comments raised in a previous round of review and you feel that this manuscript is now acceptable for publication, you may indicate that here to bypass the “Comments to the Author” section, enter your conflict of interest statement in the “Confidential to Editor” section, and submit your "Accept" recommendation.

Reviewer #2: (No Response)

Reviewer #3: All comments have been addressed

2. Is the manuscript technically sound, and do the data support the conclusions?

Reviewer #2: Yes

Reviewer #3: Yes

3. Has the statistical analysis been performed appropriately and rigorously? 

Reviewer #2: Yes

Reviewer #3: Yes

4. Have the authors made all data underlying the findings in their manuscript fully available?

Reviewer #2: Yes

Reviewer #3: Yes

5. Is the manuscript presented in an intelligible fashion and written in standard English?

Reviewer #2: Yes

Reviewer #3: Yes

6. Review Comments to the Author

Reviewer #2: Responses to the manuscript R1.

Regarding #1.

I understand the rational for participant classification regarding marital status. Thank you so much. By the way, in the "additional information" (L152-), the number of persons (percentages) for each classes in married men and women were identical. Please confirm it.

Regarding #3.

Thank you for explaining regarding "whether all the independent variables were entered simultaneously or not in multivariate analysis". Since this is very important information for interpreting findings, the authors should clearly state the information in the text body (in the statistical analyses section), I believe.

Reviewer #3: This study analyzed the changes in subjective well-being over time and the factors associated with this change. The finding will be useful for policy-making to promote subjective well-being. In the current manuscript, author has addressed my all comments.

7. PLOS authors have the option to publish the peer review history of their article (what does this mean?). If published, this will include your full peer review and any attached files.

Reviewer #2: No

Reviewer #3: No

---

## [Author Response · Author response to Decision Letter 1]

6 Jul 2023

Response to Reviewers

To Reviewer #2:

We sincerely thank you again for the thoughtful and constructive feedback you provided regarding our manuscript, PONE-D-23-04435R1：Factors Related to Subjective Well-being Among Community-Dwelling Older Adults Living Alone: A Stratified Analysis by Sex and Marital Status from the JAGES.

Regarding #1.

I understand the rational for participant classification regarding marital status. Thank you so much. By the way, in the "additional information" (L152-), the number of persons (percentages) for each classes in married men and women were identical. Please confirm it. 

⇒We apologize for our typo and thank you for your careful peer review. We have made the following corrections：　

 　Additional information: The married men group consisted of 1,803 men, among whom 63 (3.5%) were currently married, 1,005 (55.7%) were widowed, and 735 (40.8%) were divorced. The married women group consisted of 5,372 women, among whom 54 (1.0%) were married, 4,319 (80.4%) were widowed, and 999 (18.6%) were divorced. In addition, individuals in the “married with a spouse” group were assumed to be those whose spouses lived alone because they were moved to an institution or for other reasons.　(L.152～)

Regarding #3.

Thank you for explaining regarding "whether all the independent variables were entered simultaneously or not in multivariate analysis". Since this is very important information for interpreting findings, the authors should clearly state the information in the text body (in the statistical analyses section), I believe.

⇒Thank you for your important comments. We have added a note about a detailed description of the analytical method in statistical analyses section, as follows: Multivariate analyses were not performed for each independent variable, because of multicollinearity. Univariate logistic regression analyses were performed for each independent variable, and in each univariate logistic regression analysis, gender and SWB level at baseline were entered and adjusted to determine whether the 3-year change in SWB was associated with independent functional factors and interpersonal relationship factors, adjusting for age and SWB level at baseline. (L.231～)

Reviewer #3: This study analyzed the changes in subjective well-being over time and the factors associated with this change. The finding will be useful for policy-making to promote subjective well-being. In the current manuscript, author has addressed my all comments.

⇒Thank you again for the constructive feedback.

---

## [Editor Report · Decision Letter 2]

21 Jul 2023

Factors Related to Subjective Well-being Among Community-Dwelling Older Adults Living Alone: A Stratified Analysis by Sex and Marital Status from the JAGES

PONE-D-23-04435R2

Dear Dr. Tadaka,

We’re pleased to inform you that your manuscript has been judged scientifically suitable for publication and will be formally accepted for publication once it meets all outstanding technical requirements.

Kind regards,

Shaonong Dang, PhD

Academic Editor

PLOS ONE

Additional Editor Comments (optional):

Authors have addressed the concerns from the reviewers. And the manuscript has been improved much for publication.
---

## [Editor Report · Acceptance letter]

14 Aug 2023

PONE-D-23-04435R2 

Factors Related to Subjective Well-being Among Community-Dwelling Older Adults Living Alone: A Stratified Analysis by Sex and Marital Status from the JAGES 

Dear Dr. Tadaka:

I'm pleased to inform you that your manuscript has been deemed suitable for publication in PLOS ONE. Congratulations! Your manuscript is now with our production department. 

Kind regards, 

on behalf of

Dr. Shaonong Dang 

Academic Editor

PLOS ONE